# Efficacy of Fractional Laser on Steroid Receptors in GSM Patients

**DOI:** 10.3390/bioengineering10091087

**Published:** 2023-09-15

**Authors:** Stella Catunda Pinho, Thais Heinke, Paula Fernanda Santos Pallone Dutra, Andreia Carmo, Camilla Salmeron, Luciana Karoleski, Gustavo Focchi, Neila Maria Góis Speck, Beatrice Marina Pennati, Ivaldo Silva

**Affiliations:** 1Department of Gynecology, Paulista Medical School, Federal University of Sao Paulo, UNIFESP/EPM, Sao Paulo 04021-001, Brazil; stellacatunda@icloud.com (S.C.P.); thaish1978@gmail.com (T.H.); paula.pallone@yahoo.com.br (P.F.S.P.D.); carmoandreia@gmail.com (A.C.); camillacparente@hotmail.com (C.S.); lukarol@hotmail.com (L.K.); gustavo.focchi@terra.com.br (G.F.); nezespeck@uol.com.br (N.M.G.S.); ivaldo.silva@gmail.com (I.S.); 2Clinical Research and Practice Department, El.En. Group, 50041 Calenzano, Italy; 3Department of Education, ICT and Learning, Østfold University College, 1757 Halden, Norway

**Keywords:** laser therapy, genitourinary syndrome of menopause, hormone therapy, menopause, estrogen

## Abstract

Background: To compare the efficacy of CO_2_ fractional laser with that of topical estriol for treating genitourinary syndrome of menopause and to investigate the relationship between epithelial thickness and vaginal atrophy. Methods: Twenty-five menopausal women were randomized to receive either laser or estrogen treatment. Vaginal biopsies before and after treatment were compared to assess the amount and distribution of estrogen and progesterone receptors. Results: Estrogen receptor levels were statistically similar between groups before and after treatment. Although there was no change over time in the estrogen group, an increase in receptor levels was confirmed in the laser group. Changes in estrogen receptor levels showed no association with treatment. Progesterone receptor levels were statistically similar between groups throughout treatment. There was no change over time in both groups. These changes displayed no association with the type of treatment. There was no significant correlation between epithelium thickness and estrogen or progesterone receptor levels. Conclusions: Estrogen and progesterone receptor levels increased and were maintained, respectively, in the vaginal epithelium in both groups. There was no significant relationship between epithelium thickness and receptor density. Laser therapy had similar outcomes to the gold standard without involving the disadvantages of hormone therapy.

## 1. Introduction

Because of socioeconomic and sociocultural changes, a significant increase in life expectancy has occurred, especially among women. Accordingly, the number of women reaching menopause increases annually [1].

Menopause is a period of physiological changes that affect various organs and systems, thereby compromising the quality of life for many women [1]. It is marked by ovarian failure and decreased estrogen production, which can generate physical and neurovegetative symptoms [2]. Menopausal women are affected by symptoms involving the genital tract that are collectively referred to as genitourinary syndrome of menopause (GSM). This syndrome describes various symptoms and signs of menopause associated with physical and hormonal changes in the vulva, vagina, and lower urinary tract. GSM also includes sexual symptoms and tends to worsen progressively over time [3].

The lower genital and urinary tracts have the same embryonic origin and develop in anatomically closed sites. The presence of estrogen receptors in these tissues is related to their trophism, as estrogen stimulates cell growth in these structures [4]. 

Several steroid hormones (androgens, estrogens, and progestogens) have specific effects on the vaginal epithelium. These hormones are synthesized from cholesterol in various endocrine tissues. They are transported to their target cells through the bloodstream via transport proteins, where they bind to nuclear receptors and modify the expression of specific genes, affecting the development of the vaginal epithelium and also influencing a variety of other female reproductive functions [5]. 

The vagina expresses alpha and beta estrogen receptors found in the nucleus, where they can stimulate gene transcription to control cellular activities. Therefore, the administration of conjugated estrogens to menopausal women leads to intense epithelial proliferation, and their smears will comprise numerous, mainly superficial, epithelial cells [6,7]. Progesterone is associated with considerable proliferation of the intermediate layer, which is intensely loaded with glycogen. The parabasal and basal cells decrease and disappear; gradually, they are renewed by intermediate cells [7]. 

The first genital manifestation is atrophy of the vagina, characterized by thinning of the epithelium, loss of roughness, pallor of the mucosa, and reduced vaginal secretion. The ageing process contributes to sharper manifestations of these symptoms [8]. 

With GSM, the alteration of the collagen fibers, vascularization, decreased estrogen receptors, and thinning of the vaginal epithelium occur. These changes make genitourinary structures more susceptible to trauma, infections, and bleeding, leading to extreme discomfort [6,7,8,9]. The goal of GSM treatment is to relieve symptoms and improve the quality of life. The preferred treatment is systemic estrogen-based hormone therapy, which results in better outcomes, as proven by several studies [5,6]. However, there are side effects and contraindications [5]. Therefore, alternative non-hormonal therapies are utilized to relieve this condition [6,10,11,12,13]. Additionally, the utilization of light amplification by stimulated emission of radiation (laser) for treating GSM is now increasing [13]. The laser’s energy acts on tissues in a controlled manner; it has a biochemical effect, capable of stimulating the release of serotonin, histamine, and bradykinin [8,9]. It also generates a thermal effect on vaginal tissues, thus creating an environment similar to that of the necrotic–inflammatory scarring process by producing interleukins and growth factors that further stimulate and enable the migration of fibroblasts, producing collagen and the extracellular matrix [10,11]. The use of a laser has no contraindications and is safe and painless. Despite a few studies on laser treatment, it is now commonly used, and research shows that it provides relief from symptoms [6,10]. 

We aimed to analyze the efficacy of the CO_2_ fractional laser and compare its use with that of topical estriol based on the amount and distribution of estrogen and progesterone receptors in the vaginal epithelium. We also aimed to analyze its correlation with epithelial thickness in GSM patients. Our specific objectives were to document whether vaginal laser treatment can alter the amount of estrogen and progesterone receptors in the vaginal epithelium using immunohistochemical analysis and to determine if there is any correlation between the thickness of the vaginal epithelium and the estrogen and progesterone receptors.

## 2. Materials and Methods

### 2.1. Study Design

This was a prospective, randomized clinical study. The selection criteria included menopause with at least 1 year of amenorrhea and attendance at the lower genital tract pathology outpatient clinic at our institution. 

### 2.2. Sample

Convenience sampling was performed and calculated based on the spontaneous flow of patients from the outpatient clinic during a 12-month period. Randomization was applied using a website (https://www.random.org/, last accessed on 26 August 2020) accessed at the time of the patient’s arrival at the clinic. Twenty-seven patients were selected. However, two were excluded because of changes in their oncotic cytology results, and another one dropped out of the study. Of the 25 patients, 13 and 12 comprised the laser application group (Group L) and the estrogen group (Group E), respectively. Group E underwent topical estrogen therapy (1 mg/g estriol cream), while Group L underwent treatment with the CO_2_ fractional laser (MonaLisa Touch, created by Deka, an Italian manufacturer in Florence). The CO_2_ laser beam was provided using a vaginal probe (V2LR scanner with a 360° vaginal probe) inserted up to the top of the vaginal introitus. At each scan, the vaginal probe was withdrawn to provide a complete treatment of the vaginal wall. All patients underwent anamnesis, gynecological examination, pelvic–vaginal oncotic cytology, hormonal cytology, and the measurement of serum estradiol levels before treatment. Patients in Group E underwent a biopsy of the proximal third of the right vaginal wall; then, they were instructed to apply vaginal cream with estriol. Patients in Group L underwent a similar biopsy; then, they underwent three sessions of fractional laser application. Both groups underwent another biopsy of the right vaginal wall 30 days after the respective treatments. The patients were instructed to complete an informed consent form that included information and guidance regarding the biopsy, the application of the laser or use of vaginal estriol, and care after treatment.

### 2.3. Eligibility Criteria

#### 2.3.1. Inclusion Criteria

Age 40–65 years;Absence of menstruation for at least 12 months;Presence of symptoms attributable to GSM (i.e., vaginal dryness/dyspareunia);Women who underwent a conventional colposcopic examination and a Frost index consistent with atrophy;Women who had not undergone hormone replacement in the past 12 months.

#### 2.3.2. Exclusion Criteria

Acute or recurrent urogenital infection or inflammation;Preneoplastic lesions;Pelvic prolapse;Pregnancy;Immunosuppression;Severe chronic disease;Previous pelvic surgery;Use of hormone therapy for <1 year;Previous pelvic radiotherapy.

### 2.4. Laser Application

The patient was placed in the gynecological position, and the standard probe of the MonaLisa Touch 360º device that was coupled to the laser scanner was inserted. The operator wore standardized protective eyewear in accordance with the manufacturer’s recommendations. The laser had a power of 40 W, with stacking of 1 and spacing of 1000 µm, and was applied through the vagina until the bottom of the vaginal sac. The entire length of the vaginal wall was scanned. The procedure, which lasted an average of 10 min, was painless and did not require anesthetics. Use of vaginal estriol: Patients were instructed to apply 1 g of estriol-based vaginal cream every night for 30 days and then weekly thereafter for 3 months. 

### 2.5. Biopsy

A biopsy of the proximal third of the right vaginal wall was performed with a Medina clamp. The samples were fixed with 10% formaldehyde, dehydrated in increasing concentrations of ethanol, diaphanized by xylol, and embedded in paraffin by an automatic tissue processor in accordance with the protocol of the Department of Pathology, University of Sao Paulo. The samples were obtained before and after both treatments by the same examiner. After fixation, paraffin blocks were sectioned with an American Optical Minot microtome (Model 820, American Optical Corporation). The 4 mm-thick sections were mounted on glass slides and stained with hematoxylin–eosin in accordance with Michalany’s technique [12,13,14]. The epithelial thickness was measured with a Breslow ruler using an optical microscope. To reduce the chance of biases of measurements, all the measurements were performed by the same person, a pathologist with more than 20 years of experience. Ten random samples were measured by the same pathologist twice, and the reliability was greater than 95% (ICC = 0.96). 

The immunohistochemical analysis of the estrogen and progesterone receptors was performed by a single pathologist. The increase in receptor expression was evaluated as follows: (i) levels were considered increased when the reading went from negative to positive (lower layer, up to the middle third, up to the upper third); (ii) when the reading went from positive in each lower layer to positive up to the middle third or up to the upper third; or (iii) when the reading went from positive up to the medium third and up to the upper third layer. Levels were considered decreased when the direction of the change was opposite that described here.

### 2.6. Immunohistochemistry

Automated procedure: An automated procedure was used for staining estrogen and progesterone antibodies. Special immunohistochemical staining was performed on polarized slides with 4 µm histological sections obtained by standardized conventional microtomy. The sections were dewaxed in an oven (57 °C, 60 min). Then, the automated protocol was applied using the Autostainer 48 (DAKO, Glostrup, Denmark) according to the manufacturer’s guidelines and using the following steps: (i) antigenic recovery in PT-LINK buffer solution (DAKO); (ii) washing in buffer solution for 5 min; (iii) application of the primary antibody estrogen receptor clone EP1 and progesterone receptor (clone PgR 636) for 20 min; (iv) amplification of the reaction with the EnVision Flex system (DAKO) for 20 min; (v) washing in buffer solution for 5 min; (vi) development of the reaction in diaminobenzene tetrahydrochloride (DAB) (Sigma-Aldrich Chemical, St. Louis, MO, USA); (vii) flex for 5 min; (viii) counterstaining with hematoxylin flex for 10 min. Special immunohistochemical staining was performed on polarized slides with 4 µm histological sections obtained using the standardized conventional microtomy technique; these slides were submitted to dewaxing in an oven set at 57 °C for 60 min. Then, the slides were cooled in a PT-LINK buffer solution (DAKO) until they reached 20 °C and washed in the same solution. Next, endogenous peroxidase was blocked with two washes for 10 min each in 20 volumes of hydrogen peroxide, followed by washing in running water and immersion in buffer solution. The slides were immersed in a protein-blocking serum for 10 min (Novocastra/K3468), washed, and immersed in a buffer solution. The primary antibody was applied overnight in a humid chamber at a controlled temperature (3 °C), followed by washing in a buffer solution using the Novolink system (Novocastra RE 7280). Finally, the reaction was developed in DAB for 5 min and counterstained using Harris hematoxylin for 3 min. 

### 2.7. Statistical Analyses

Statistical analyses were initially performed using the mean, median, minimum, and maximum values, standard deviation, and absolute and relative frequencies (percentage), as well as individual profile plots (lines) and one-dimensional, two-dimensional, and bar dispersion. The following inferential analyses were used to confirm or refute evidence from the descriptive analysis: Student’s *t*-test was used for independent samples [15] to compare the groups (estrogen and laser) based on the current age, body mass index (BMI), number of normal deliveries, and increased thickness of the vaginal epithelium;Pearson’s chi-squared and Fisher’s exact test or its extension [16] were used to compare groups (estrogen and laser) based on marital status, schooling, hypertension, osteoporosis, depression, increased thickness of the vaginal epithelium, estrogen levels, and progesterone receptor levels;The Mann–Whitney U-test [17] was used to compare groups (estrogen and laser) based on age at the beginning of menopause;The Wilcoxon rank-sum test [17] was used to compare levels (scores) of estrogen and progesterone receptors between time periods (initial and final);A repeated-measures analysis of variance [18] was used to compare the thickness of the vaginal epithelium of the groups (estrogen and laser) and time periods (initial and final);Spearman’s correlation coefficient [17] was used to quantify the correlation of the vaginal epithelial thickness with estrogen and progesterone receptor levels. An alpha significance level of 5% was used for all conclusions obtained through inferential analyses. The data were analyzed using Excel 2010 for Windows (Microsoft, Redmond, WA, USA) for proper storage of the information. Statistical analyses were performed using SPSS Statistics Version 24 (IBM, Armonk, NY, USA) and R Version 3.6.3 (R Foundation, Vienna, Austria) [19].

## 3. Results

Twelve women (48.0%) were allocated to Group E and thirteen (52.0%) to Group L (Table 1). In Group E, the mean age was 55.5 years (range: 50–62 years), and the mean body mass index (BMI) was 27.3 kg/m^2^ (range: 22.0–33.7 kg/m^2^). These women entered menopause at a mean age of 44.5 years (range: 30–51 years). In Group L, the mean age was 55.8 years (range: 50–63 years), and the mean BMI was 28.2 kg/m^2^ (range: 22.1–37.9 kg/m^2^). Five (38.5%) had hypertension; most were married (84.6%); approximately seven (53.8%) had completed high school. These women entered menopause at a mean age of 47.8 years (range: 41–54 years). 

In Group E, estrogen receptor levels increased in seven (58.3%), decreased in three (25.0%), and did not change in two (16.7%) patients, whereas in Group L, ten (76.9%) patients had increased levels and three (23.1%) had unchanged levels (Table 2 and Figure 1).

Progesterone receptor levels increased in two (16.7%) patients and did not change in ten (83.3%) patients in Group E. In Group L, two (15.4%) patients had increased levels, one (7.7%) patient had decreased levels, and ten (76.9%) patients had no change in their levels (Table 2 and Figure 2).

Estrogen receptor levels were statistically similar between groups before (*p* = 0.769) and after treatment (*p* = 0.586). There was no change over time in Group E (*p* = 0.207), but an increase was confirmed in Group L (*p* = 0.004). Nevertheless, changes in estrogen receptor levels displayed no association with treatment (*p* = 0.238). Progesterone receptor analyses indicated that the levels of this receptor were statistically similar between groups at treatment initiation (*p* = 0.124) and after treatment (*p* > 0.999), and there was no change over time in either group (Group E, *p* = 0.157; Group L, *p* = 0.564). Changes in progesterone receptor levels displayed no association with treatment (*p* > 0.999). No significant correlation was found between epithelial thickness and estrogen and progesterone receptor levels (Table 3). 

## 4. Discussion

This study revealed an increase in estrogen receptor levels and the maintenance of progesterone receptor levels in the vaginal epithelium in both groups, with no statistical difference between them. There was no relationship between the area of expression (thickness) and the density of these receptors.

The increase in life expectancy of women necessitates further scientific study and the development of appropriate therapies for the relief of GSM symptoms that can be used for all women [20,21,22]. GSM is a factor that impacts the quality of life and affects a broad age group [1,3,4,6]. Changes resulting from low postmenopausal estrogen levels immensely affect the female genitourinary tract, causing signs and symptoms characteristic of this phase, which tends to worsen over time. Topical estradiol therapy is the gold standard treatment for this condition because of the contraindications associated with or fear of the use of other therapies.

Unfortunately, this does not satisfy all patients [6,7,14,23,24,25,26]. Several studies [2,4,7,8,10,25,26,27,28] searching for non-hormonal therapies for the treatment of GSM are still undergoing development and attempting to determine the efficacy and safety of the vaginal CO_2_ laser. Some studies have shown relief from symptoms and improvement in sexual health for these patients [18,20].

The vagina expresses the nuclear estrogen (alpha and beta) and progesterone receptors, which can bind directly to cellular DNA and regulate the expression of target genes [12,29,30]. There have been no published clinical studies demonstrating the impact of laser on the estrogen and progesterone receptors in the vaginal epithelium. This study is the first to investigate the repercussions of this therapy at the level of these structures and to show an increased number of estrogen receptors. In contrast, the levels of progesterone receptors in the vaginal epithelium in both groups remained unchanged. However, no statistical difference was found between the groups. Furthermore, there was no difference in the density of receptors according to the measured thickness. Hence, we believe that possible changes in density can be demonstrated in future studies with more participants. Another possible reflection is that estrogen known to be present in greater quantities during menopause was bound to the increased receptors in Group L, thus improving symptoms and, thereby, indicating the possibility of a non-hormonal method with relevant effects on these structures, with the same benefits as the current gold standard. The evaluation of sexual function is complex and multifactorial, and it does not depend exclusively on vaginal trophism [15,17,18]. We can infer that vaginal laser therapy has a positive impact on sexual function that is not solely related to vulvovaginal atrophy. Therefore, we believe that improvements in these parameters could also be attributed to the local action on increased receptors. Therefore, the vaginal laser is an excellent option and an important non-hormonal tool that can improve the quality of life, despite its higher cost. Further research is still needed to evaluate the short- and long-term effects of this technique. It is not yet possible to predict the duration of changes; hence, prospective complementary studies with more participants and a longer duration are required. The possibility of increasing the population size for further long-term evaluations of the parameters should not be ruled out. Although the principles of laser therapy are established in medicine, especially in dermatology, this study provides new and unprecedented therapeutic perspectives regarding the application of laser technology in gynecology because it is free from side effects and possible endometrial repercussions caused by using hormones [21,22]. Currently, a better understanding of the duration of the effects of this method and its long-term consequences and the functions of these processes at the epithelial, cellular, and nuclear levels is necessary.

## 5. Conclusions 

This study revealed an increase in estrogen receptor levels and the maintenance of progesterone receptor levels in the vaginal epithelium in both groups, with no statistical difference between them. There was no relationship between the thickness and density of these receptors; however, the results should be proven by future studies with a larger number of participants. This study showed that it is possible to obtain good results with a therapy that, although more expensive and less accessible, does not involve the disadvantages of the regular, topical application of estriol and the known contraindications of hormone therapy. Further studies are required before laser therapy can be used routinely and safely to treat postmenopausal genitourinary syndrome and other similar conditions

## Figures and Tables

**Figure 1 bioengineering-10-01087-f001:**
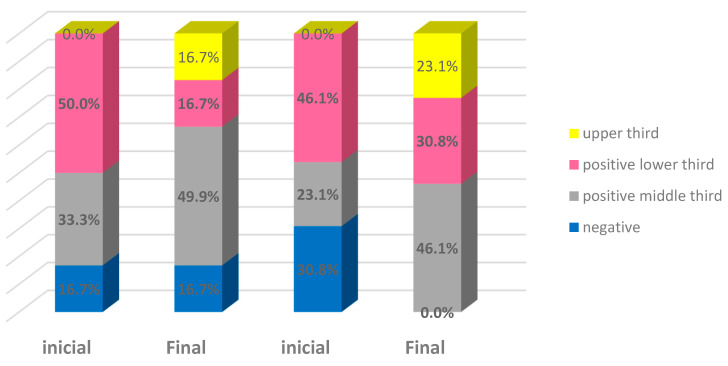
Distribution of estrogen receptor levels in the epithelium of patients according to time and group. Estrogen group: blue—negative; grey—positive middle third. Laser group: pink—positive lower third; yellow—upper third.

**Figure 2 bioengineering-10-01087-f002:**
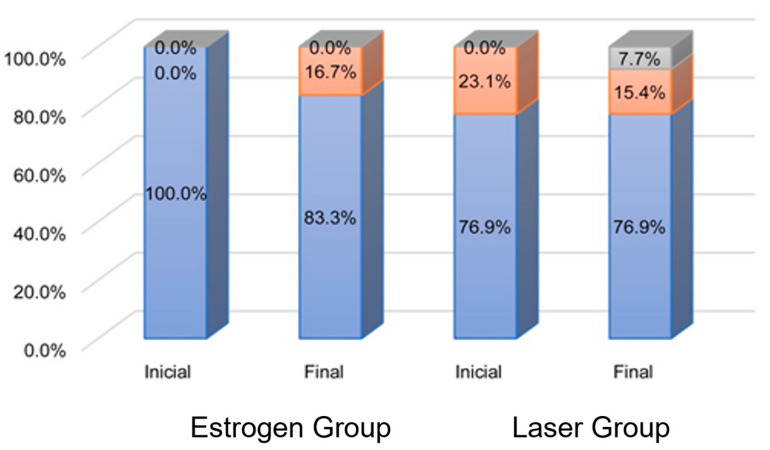
Distribution of progesterone receptor levels in the epithelium of patients according to time and group. Blue: negative; pink: positive lower third; gray: positive middle third.

**Table 1 bioengineering-10-01087-t001:** General characteristics of women according to group. ^a^ Student’s *t*-test for independent samples. ^b^ Mann–Whitney test.

		Group						
		Estrogen		Laser		Total		
		(*n* = 12)		(*n* = 13)		(*n* = 25)		*p*
Current age (years)	Average	55.5		55.8		55.6		0.880 ^a^
	Median	55.0		55.0		55.0		
	Minimum	50.0		50.0		50.0		
	Maximum	62.0		63.0		63.0		
	Standard deviation	4.4		4.5		4.3		
Body mass index (kg/m^2^)	Average	27.3		28.2		27.8		0.591 ^a^
	Median	26.7		26.5		26.6		
	Minimum	22.0		22.1		22.0		
	Maximum	33.7		37.9		37.9		
	Standard deviation	3.4		4.7		4.1		
Number of normal deliveries	0	3	25.0%	2	15.4%	3	25.0%	2
	1	-	-	1	7.7%	-	-	1
	2	5	41.7%	5	38.5%	5	41.7%	5
	3	2	16.7%	4	30.8%	2	16.7%	4
	4	1	8.3%	1	7.7%	1	8.3%	1
	6	1	8.3%	-	-	1	8.3%	-
Marital status	Married	10	83.3%	11	84.6%	10	83.3%	11
	Single	-	-	2	15.4%	-	-	2
	Widowed	2	16.7%	-	-	2	16.7%	-
Schooling	Elementary incomplete	3	25.0%	4	30.8%	3	25.0%	4
	Elementary complete	3	25.0%	-	-	3	25.0%	-
	High school complete	5	41.7%	7	53.8%	5	41.7%	7
	Higher education complete	1	8.3%	2	15.4%	1	8.3%	2
Systemic arterial hypertension	Yes	6	50.0%	5	38.5%	6	50.0%	5
	No	6	50.0%	8	61.5%	6	50.0%	8
Osteoporosis	Yes	-	-	2	15.4%	-	-	2
	No	12	100.0%	11	84.6%	12	100.0%	11
Depression	Yes	4	33.3%	2	15.4%	4	33.3%	2
	No	8	66.7%	11	84.6%	8	66.7%	11
Age at the onset of menopause (years)	Average	44.5		47.8		46.2		0.366 ^b^
	Median	47.5		49.0		49.0		
	Minimum	30.0		41.0		30.0		
	Maximum	51.0		54.0		54.0		
	Standard deviation	7.0		4.0		5.8		

**Table 2 bioengineering-10-01087-t002:** Distribution of estrogen and progesterone receptor levels in the epithelium of patients over time and according to the group.

	Estrogen Group				Laser Group				Total			
	Initial		End		Initial		End		Initial		End	
Estrogen receptor												
Negative (score of 0)	2	16.7%	2	16.7%	4	30.8%	-	-	6	24.0%	2	8.0%
Positive in lower layer (score of 1)	6	50.0%	2	16.7%	6	46.2%	4	30.8%	12	48.0%	6	24.0%
Positive up to the middle third (score of 2)	4	33.3%	6	50.0%	3	23.1%	6	46.2%	7	28.0%	12	48.0%
Positive up to the upper third (score of 3)	-	-	2	16.7%	-	-	3	23.1%	-	-	5	20.0%
Total	12	100.0%	12	100.0%	13	100.0%	13	100.0%	25	100.0%	25	100.0%
Progesterone receptor												
Negative (score of 0)	12	100.0%	10	83.3%	10	76.9%	10	76.9%	22	88.0%	20	80.0%
Positive in lower layer (score of 1)	-	-	2	16.7%	3	23.1%	2	15.4%	3	12.0%	4	16.0%
Positive up to the middle third (score of 2)	-	-	-	-	-	-	1	7.7%	-	-	1	4.0%

**Table 3 bioengineering-10-01087-t003:** Estimation of Spearman’s correlation coefficient (S) for vaginal epithelium thickness and estrogen and progesterone receptor levels according to time and group.

		Estrogen Group		
Time	Correlated pairs	S *	CI **	*p*
Initial	Thickness x estrogen	0.118	−0.489; 0.648	0.715
	Thickness x progesterone	-	-	-
Final	Thickness x estrogen	−0.171	−0.647; 0.490	0.596
	Thickness x progesterone	0.468	−0.145; 0.821	0.125
		Laser group		
Time	Correlated pairs	S *	CI **	*p*
Initial	Thickness x estrogen	0.217	−0.379; 0.689	0.477
	Thickness x progesterone	−0.100	−0.617; 0.477	0.745
Final	Thickness x estrogen	0.331	−0.269; 0.746	0.270
	Thickness x progesterone	−0.203	−0.678; 0.392	0.506
		Estrogen + laser		
Time	Correlated pairs	S *	CI **	*p*
Initial	Thickness x estrogen	0.208	−0.089; 0.633	0.317
	Thickness x progesterone	−0.140	−0.507; 0.270	0.504
Final	Thickness x estrogen	0.150	−0.261; 0.515	0.475
	Thickness x progesterone	−0.116	−0.489; 0.293	0.581

* Spearman’s correlation coefficient. ** The 95% confidence interval for Spearman’s correlation coefficient. CI, confidence interval.

## Data Availability

Data that support the study findings are available upon request from the corresponding author.

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
