# Peer review of "Efficacy of Fractional Laser on Steroid Receptors in GSM Patients"

_bioengineering, 2023, doi:10.3390/bioengineering10091087_

Round 1

Reviewer 1 Report

Overall the manuscript appears to be clearly and carefully written. I think that the manuscript might deserve publication in the Journal of Bioengineering after some points are dealt with and some missing details are added prior to publication as follows:

1-      Please add to the text some details about the laser system used in the present study.

2-      The authors should state why they used a CO2 laser.

3-      What effects should be expected if laser power was varied?

4-      What effects should be expected if the irradiation time was varied?

5-      Nothing is mentioned in the text about the uncertainty in the measurements, so please discuss this issue in the text.

6-      The authors should describe how their work is unique and how it differs from previous studies.

Overall the manuscript appears to be clearly and carefully written.

Author Response

1-     Please add to the text some details about the laser system used in the present study.

Thank you for the suggestion. An extra description of the laser system was added to the manuscript.

2-      The authors should state why they used a CO2 laser.

There are in the literature the confirmation its efficacy in changing and rejuvenating vulvovaginal tissue in patient affected by VVA. Subsequent studies correlated this genital remodeling to vaginal atrophy symptoms improvements and also, we were able to have the machine in our outpatients clinic

3-      What effects should be expected if laser power was varied?

See the answer to question #4

4-      What effects should be expected if the irradiation time was varied?

Thank you for the questions. The energy value is proportionally related to power and dwell time. This way, the higher the energy is, the higher the thermal and ablation effect will be. This essential matter was already demonstrated in 1999 by Ross et al (DOI: 10.1001/archderm.135.4.444).

5-      Nothing is mentioned in the text about the uncertainty in the measurements, so please discuss this issue in the text.

Thank you for the comment. The text below was added to the manuscript.

“To reduce the chance of biases of measurements, all the measurements were performed by the same person, a pathologist with more than 20 years of experience. Ten random samples were measured by the same pathologist twice and the reliability was superior than 95 % (ICC =0.96).”

6-      The authors should describe how their work is unique and how it differs from previous studies.

This is the first study revealed an increase in estrogen receptor levels and the maintenance of progesterone receptor levels in the vaginal epithelium using CO2 laser. This study shows it is possible to obtain good results without using topical hormone.

Reviewer 2 Report

Dear colleagues,
In this manuscript, the authors compare the efficacy of fractional laser with that of topical estriol for treating genitourinary syndrome of menopause. The study is prospective, randomized clinical study (n=25). The results are interesting. The figures reflect the results of the study. Despite the very impression of the article, there are some questions which could improve the article in my opinion, partly:

The abstract needs stylistic correction in the beginning part especially; digital results should be added also.

The discussion could be improved with more sources devoted to hormonal regulation of the female reproductive system.

Conclusions could be improved with digital data of obtained results.

The article needs stylistic correction for better connection between some sentences and parts. 

In summary, I have been satisfied with the high level of the article. I believe this manuscript will attract significant attention from the research community. In my personal opinion, the article is very valuable, a great prospect for further research, and, after minor corrections, can be recommended for publication. 

Dear colleagues,
The abstract needs stylistic correction in the beginning part especially; digital results should be added also.

The article needs stylistic correction for better connection between some sentences and parts. 

Author Response

Thank you for you comments and suggestions. The manuscript was modified according to your advice.

Round 2

Reviewer 1 Report

The authors have made reasonable changes to the manuscript in response to my previous suggestions and concerns. In my opinion, the manuscript now has all information and is ready for publication as a regular article in the Journal " Bioengineering ".

Moderate editing of English language required